# Structural Equation Modelling as a Proof-of-Concept Tool for Mediation Mechanisms Between Topical Antibiotic Prophylaxis and Six Types of Blood Stream Infection Among ICU Patients

**DOI:** 10.3390/antibiotics13111096

**Published:** 2024-11-18

**Authors:** James Hurley

**Affiliations:** 1Melbourne Medical School, University of Melbourne, Parkville, VIC 3052, Australia; hurleyjc@unimelb.edu.au; 2Ballarat Health Services, Grampians Health, Ballarat, VIC 3350, Australia; jamesh@gh.org.au; 3Ballarat Clinical School, Deakin University, Ballarat, VIC 3350, Australia

**Keywords:** bacteremia, *Pseudomonas aeruginosa*, *Staphylococcus aureus*, *Acinetobacter*, structural equation modelling, gut/blood microbiome, *Candida*, intensive care

## Abstract

Whether exposing the microbiome to antibiotics decreases or increases the risk of blood stream infection with *Pseudomonas aeruginosa*, *Staphylococcus aureus*, *Acinetobacter*, and *Candida* among ICU patients, and how this altered risk might be mediated, are critical research questions. Addressing these questions through the direct study of specific constituents within the microbiome would be difficult. An alternative tool for addressing these research questions is structural equation modelling (SEM). SEM enables competing theoretical causation networks to be tested ‘en bloc’ by confrontation with data derived from the literature. These causation models have three conceptual steps: exposure to specific antimicrobials are the key drivers, clinically relevant infection end points are the measurable observables, and the activity of key microbiome constituents on microbial invasion serve as mediators. These mediators, whether serving to promote, to impede, or neither, are typically unobservable and appear as latent variables in each model. SEM methods enable comparisons through confronting the three competing models, each versus clinically derived data with the various exposures, such as topical or parenteral antibiotic prophylaxis, factorized in each model. *Candida* colonization, represented as a latent variable, and concurrency are consistent promoters of all types of blood stream infection, and emerge as harmful mediators.

## 1. Introduction

The human microbiome, its relationship to human health, and the specific impact of antimicrobials are of great interest [1,2]. There is a long-standing suspicion that antimicrobial exposure changes the microbiome to influence various aspects of human health [3]. This suspicion is especially strong in the context of critical illness given the large amount of antimicrobial exposure occurring in this context [4,5].

Stated simply as research questions: Does antimicrobial exposure, given as prophylaxis, increase or decrease the risk of blood stream infection among ICU patients? Does this impact vary for different types of blood stream infections? More fundamentally, does the microbiome mediate this risk, and if so, by what mechanism?

A conventional approach to address these research questions would face multiple challenges. Which are the relevant microbiome constituents? What are the postulated mechanisms of mediation? Which specific antimicrobials are critical, and what is their impact? Is it possible to account for the other possible drivers of this mediation, such as exposures to probiotics or non-antimicrobial interventions? In addressing these questions, is there an alternative approach?

Structural equation modelling (SEM) is a suite of multivariable statistical techniques which offers an alternative approach to addressing these research questions [6]. SEM enables causal relationships within structural models to be tested by confrontation against data. Hence, in considering SEM as an alternative approach, we need to consider the three competing hypotheses of the relationship between exposure of the microbiome to antimicrobial prophylaxis and the risk of blood stream infection (Figure 1), being colonization resistance and its counterpart, colonization susceptibility, and a third concept of antimicrobial action directly controlling gut overgrowth (COGO) [7,8,9,10,11,12,13,14,15].

For the research questions, the clinical end point of most interest will be blood stream infection. This has the key advantage of being well defined and clinically relevant. Other end points, such as pneumonia, which may be less well defined or whose clinical relevance is uncertain, may also provide information indirectly relevant to the research question [10,11,12,13,14,15,16,17,18,19,20]. In studying blood stream infections, it is necessary to distinguish among the separate contributions of the various components of the commonly studied antibiotic prophylaxis regimens, those being the topical antibiotic, parenteral antibiotic, and the antifungal components. The SEM approach enables this.

## 2. Competing Hypotheses

The research into antimicrobials and other methods for preventing acquired infections among ICU patients extends over nearly fifty years and has generated >300 controlled trials of various antimicrobial-based [21,22,23,24,25,26,27,28] and also non-antimicrobial-based interventions [29,30,31,32], which have been summarized in systematic reviews [17,18,19,20,21,22,23,24,25,26,27,28,29,30,31,32]. Methods based on the use of topical antibiotic prophylaxis, with or without PPAP as an additional component, appear to be the most effective infection prevention method, and there are three competing models of how this antimicrobial exposure modifies the risk of blood stream infection (Figure 1 and Figure 2) [17,18,19,20]. Note that these competing models overlap, with shared mediation elements.

### 2.1. Colonization Resistance (Figure 1b)

The history of research on how microbes might provide colonization resistance to invasive infection is extensive [16]. The specific microbes within the microbiome that contribute to colonization resistance either may not be culturable or may otherwise be difficult to identify [7]. However, it is commonly presumed that antimicrobials with selective activity might act as drivers to enhance the activity of the microbes mediating colonization resistance [10,11,12].

The original preclinical studies on colonization resistance, dating from the early 1970s, generated enthusiasm that within the normal gastrointestinal tract flora were specific microbes that served to block infections arising from pathogenic bacteria there [9]. Exposing neutropenic mice to a combination of streptomycin and neomycin enhanced a state of ‘colonization resistance’ within the mice. This enhanced colonization resistance resulting from the selective antibiotic exposure was then transferable between mice in the same cage and also to new mice introduced to the cage which previously housed mice whose colonization resistance had been enhanced. While the transmissibility of colonization resistance implicated a component of the mouse microbiome, that microbiome component has never been identified. These mouse protection studies raised hopes for preventing infections in high-risk patient groups by using specific antimicrobial agents as prophylaxis whose spectrum of activity would promote colonization resistance and protection. The concept of achieving ‘selective digestive decontamination’ (SDD) through topical antibiotic prophylaxis was born. This concept raised hopes for developing SDD regimens for clinical application for the prevention of infection in high-risk patients [8,14,16], with the first human studies being undertaken in patients who were neutropenic following high-dose cytotoxic chemotherapy [21]. Over time, clinical use of SDD has shifted from hematology patients to patients receiving or at risk of receiving prolonged mechanical ventilation (MV) in intensive care units (ICUs) [17,18,19,20]. These patients are at high risk of acquiring invasive infections causing blood stream infection and pneumonia with microbes such as *Pseudomonas aeruginosa* and *Staphylococcus aureus* as well as fungal infections such as *Candida* during their ICU stay. How much of this risk arises from the invasive ventilation leading to respiratory tract colonization or from the gastrointestinal tract colonization is uncertain [22].

An SDD regimen, which was slightly modified from the regimen used to prevent infections among hematology patients by the addition of protocolized parenteral antibiotic prophylaxis (PPAP) and antifungal agents, appears to be the most effective intervention to prevent infection, including blood stream infection, and possibly also mortality, in the ICU context versus other interventions [18,19,20,33,34,35]. This apparent effectiveness is evident in several percentage points’ difference between control and interventions groups of randomized concurrent controlled trials (RCCT) for incidences of mortality, pneumonia, and other end points [36,37,38,39,40,41,42,43,44,45,46,47,48,49,50].

Note that the various SDD regimens as currently studied are multi-component and include some or all of antifungal, topical antibiotic, and protocolized parenteral antibiotic prophylaxis (PPAP) as components. Additionally, PPAP is sometimes used in concurrent control groups of some studies [51].

Spillover of colonization resistance to concurrent patients was seen as a threat to estimating the effect of SDD within the ICU setting. The first SDD study used a non-concurrent study design intentionally to minimize any spillover effect that might benefit concurrent control group patients [36]. Paradoxically, in several studies since, wherein control (non-concurrent) patients are not within the same ICU as the intervention patients receiving the SDD intervention, as in the three largest studies of SDD to date, the prevention effect is either less evident or absent [37,38,39]. Another paradoxical finding is that the event rates observed in control groups of RCCT studies are unusually high compared to externally derived benchmarks in comparable populations [35,40,41,42,43]. There are other paradoxical results which raise doubt as to the transmissibility of protection between patients in the mediation of the apparent SDD effect [35,43,44,45,46,47,48,49,50,51,52]. SEM enables this transmissibility to be tested as the effect of concurrency.

### 2.2. Control of Gut Overgrowth (Figure 1a)

Despite fifty years and over sixty controlled trials of SDD, the underlying mechanism of colonization resistance and even whether colonization resistance is required for SDD mediation are unclear [7,14,15]. The SDD regimens are compound, comprising several topical antimicrobials, parenteral antibiotics, and an antifungal, with numerous variations in constituents among the studied SDD regimens [18,19]. Moreover, without identification of the key microbe mediating colonization resistance, the verification of each of the three key steps in the colonization resistance model remains problematic, and the postulated chain of causation remains to be proven.

Recently, an alternative mechanism has been postulated for the apparent prevention effect of SDD wherein this intervention acts directly to mediate control of gut overgrowth (COGO) by potentially pathogenic microbes. Unlike colonization resistance, the COGO mechanism is not presumed to be transmissible between patients and ceases on termination of exposure to antibiotic prophylaxis [14,15].

COGO is of interest as a method for controlling various multi-resistant infections such as, for example, carbapenem-resistant *Enterobacteriaceae* (CRE) [53,54]. However, the evidence is mixed, and this strategy is not recommended in European guidelines [55,56]. There are concerns that this use of antimicrobials may compound the antibiotic resistance in the ICU through the population effect of antibiotics in this setting [57,58]. Moreover, SDD has ecological effects on resistant Gram-negative bacterial colonization in the ICU [59]. The failure of a recent large trial of SDD to prevent blood stream infections was attributed to the prevalence of antibiotic resistance in the ICUs of the study [38].

### 2.3. Colonization Susceptibility (Figure 1c)

In contrast to the long-studied phenomenon of colonization resistance, colonization susceptibility, wherein microbes might facilitate invasive infections, has emerged more recently as a competing hypothesis. *Candida* is a strong candidate for a microbe mediating colonization susceptibility. This is an example of an interkingdom interaction between fungi and bacteria [60,61,62,63,64,65,66,67,68,69]. There is extensive recent preclinical evidence that *Candida* facilitates invasive bacterial infection by a range of mechanisms. However, this interaction is difficult to study in the clinical context. The few small clinical studies that might bear on the question of colonization susceptibility have been small and inconclusive [70,71,72,73,74,75,76,77].

In considering *Candida* as a mediator of colonization susceptibility, by what mechanism would this promote invasive infection? At what site in the body might this interaction occur, and where is the presence of *Candida* most relevant? Does the mechanism require an increase in the level of *Candida* or merely an increase in some aspect of its activity? Beyond the mere presence of *Candida*, is there some specific functional activity of *Candida* which facilitates the interaction with other microbes to cause invasion? *Candida* is known to be transmissible in the ICU setting. Does this transmissibility implicate transmission of colonization susceptibility? In contrast to COGO, the bacterial spectrum targeted by the colonization susceptibility model should not reflect the spectrum of antimicrobial activity in the prophylaxis. Does the known spectrum of activity of antimicrobials within SDD regimens, be they antibiotics targeting various specific Gram-positive bacteria, Gram-negative bacteria, or fungi, implicate the microbial mediation underlying colonization susceptibility [77]?

Of interest, topical antibiotic prophylaxis was recently found to be a strong promoter of horizontal transmission of *Candida* colonization between mice concurrent within the same cage [78].

Given these three competing hypotheses, the effects of antibiotic prophylaxis in the ICU context are ripe for a reappraisal [79].

## 3. Lessons for Microbiome Research

Research relating to possible antimicrobial drivers of microbiome constituents towards potential clinical outcomes will encounter similar obstacles. While the conventional approach to these challenges would involve studying each step in the chain of causation separately, step by step, this is not simple. To verify a chain of causation requires each of the following to be identified: the microbial constituents central to the mediation, the most likely body site where interaction might occur, the most relevant clinical end points, and the key drivers, be they a specific class of antimicrobial or even a non-antimicrobial agent or exposure. To verify each step in this causation chain in the clinical context would be challenging. It would be ethically challenging to design a controlled trial in humans, whether using healthy volunteers or patients to address these questions, given the potential risk of harm [80]. Moreover, the potential microbiome transmissibility between patients in the ICU context poses great logistical complexity in designing a controlled study [80].

However, SEM offers an alternate approach in which a postulated chain of causation is analyzed ‘en bloc’. SEM has a long history of application in diverse subject areas of research, such as sociology and econometrics, where similar ‘chain of causation’ research questions are commonplace. SEM use within microbiome research could enable similar obstacles to likewise be circumvented to facilitate ‘chain of causation’ research questions [81,82,83,84,85].

## 4. Structural Equation Modelling (SEM)

### 4.1. SEM Origins

SEM is a family of statistical techniques which combine aspects of multiple regression and factor analysis to allow latent variables to be identified and their relationships to be evaluated. SEM enables the covariances between a set of measured outcomes within a dataset to be mapped to a small number of endogenous variables, whether observable or not (latent), which in turn are driven by the observed exogenous exposures. Usually, these relationships are modelled within a network of causation. A closely related method is that of directed acyclic graphs (DAGs). Similar techniques are applied to enable causal discovery within high-dimensional cross-sectional data derived from microbiome observations and meta-genomic data [86].

The origins of SEM can be traced to exploratory factor analysis, path analysis, and structural causal models. A feature of SEM which distinguishes it from simple linear regression is that the analytical method presumes directionality within the causal flow underlying the causation network. That is, ‘X’ is believed to mediate ‘Y’ in the cause of ‘Z’. This presumption flows from either the temporal sequence of events or prior beliefs. By contrast, tests of association using regression techniques, as widely used, have no implied causal flow or direction. That is, any association of ‘X’ with ‘Y’ and ‘Z’ is equivalent to an association of ‘Y’ and ‘Z’ with ‘X’.

In addressing research questions using SEM, the available data are used to confront several candidate networks of causation models ‘en bloc’ to find which one candidate model, allowing for parsimony, might be most consistent with the data. One commonly used information index is the Akaike information criterion (AIC), which has broad applications beyond SEM for the purpose of comparing the trade-off between model fit and model complexity. While the candidate model with the lowest AIC value is deemed optimal, this does not verify that this identified model is correct in every specific detail, but merely its being optimal amongst the range of models presented for analysis.

A modification of SEM is generalized structural equation modelling (GSEM) wherein generalized linear response functions beyond the constraint of linear response functions within standard SEM are enabled. GSEM additionally enables the inclusion of clusters with missing observations to be included in the analysis under the assumption of missing at random. For SEM analysis, the units, whether individual patients or groups of ICU patients, are at a single level. On the other hand, GSEM enables multi-level analysis combining units at different levels in the same analysis. The disadvantage of an analysis at the level of individual patients is that contextual effects caused by transmissibility cannot be estimated, as these can only be identified at the group level as a herd effect [87,88].

### 4.2. Applying SEM to Blood Stream Infection Risk

In applying SEM (or GSEM) techniques to the research questions regarding the presumed mediation of antimicrobial exposures on the risk of blood stream infections, four steps are required. Firstly, preconceived models are required which encapsulate the prior beliefs in simple outline as key defining steps. Here, the postulated models are COGO, colonization resistance, and colonization susceptibility. Notably, the various multiple components of SDD can be factorized into exposures to TAP, PPAP, and antifungal agents, to model their individual effects. This includes where the PPAP might have been used additionally in the concurrent control groups of some studies.

The second step is identifying latent variables though their relationship with measurable factors or end points. For the colonization susceptibility model, where the latent variable is *Candida* colonization, the relevant end point is the measure of *Candida* infection, being either *Candida* blood stream infection (candidemia) or *Candida* isolation from the respiratory tract. These are commonly reported among studies of infection prevention in the ICU context. Note that measurements of *Candida* colonization, measured for example as rectal colonization, are not used here. These latent variables are defined by their respective relationships with the isolation of these microbes from the blood stream and the respiratory tract, which might be considered more relevant to the model than rectal colonization.

For the colonization resistance model, the microbe remains unidentifiable. However, given the transmissibility in the mouse experiments that underlie the SDD concept, the presence of this microbe will be identifiable as a latent variable by its contribution to herd protection in patients concurrently located in the ICU (e.g., concurrent control group patients). With this model, concurrency in a group receiving antibiotic prophylaxis would be predicted to mediate a transmissible protective effect. Lastly, for COGO, the mediation of the antimicrobial exposure is directly onto the bacterial colonization leading to blood stream infection. However, there is the additional concern that bacterial colonization will rebound on cessation of COGO with withdrawal of antibiotic prophylaxis.

Thirdly, the target patient population at risk of acquiring blood stream infection against which antimicrobial exposures may have a preventive effect needs to be identified and broadly defined. Usually, a large amount of data for the end points of interest, either at the patient level or the group level of analysis drawn from this patient population, are needed. These end points would not be limited to blood stream infection and might, for example, include other end points such as pneumonia or fungal infections, that would be modified by the exposures of interest. The amount of data required varies, but several hundred ‘units’, be they patients or groups, will be required to power SEM analyses of even simple models. There is no exact formula for the minimum number of units required, but a ten-to-one ratio between ‘units’ and factors is suggested.

Finally, the models are each tested by confrontation with data from these patient populations. To the known network of causation is added these key defining steps, one at a time. The resulting models are evaluated and compared by repeating the analysis with and without each key step present in the model. From these confrontations, the optimal model emerges as the one having the lowest AIC.

I have recently used GSEM to compare the three competing models for how antimicrobial prophylaxis among ICU patients impacts the incidence of various types of blood stream infection. These models are based on the three competing models of COGO (Figure 1a), colonization resistance (Figure 1b), and colonization susceptibility (Figure 1c).

These comparisons have used data from 200 to 300 studies reporting pneumonia and blood stream infection with *Pseudomonas aeruginosa* [88], *Acinetobacter* [89], *Coagulase negative Staphylococcus* (CNS) [90], *Enterococcus* [90], *Candida* [88], and *Staphylococcus aureus* [91] in groups of ICU patients, the majority of whom received prolonged mechanical ventilation (MV). Notably, fewer than 50 of the studies reported data for more than one infection type.

In these examples, a small (three to seven) number of competing models were confronted with the clinical data. In the models, there are exposures to several key drivers other than antimicrobials, together with exposures to regimens of singleton or combination antimicrobials evaluated as infection prevention interventions. A distinct advantage of SEM methods is that the SDD regimen can be factorized into the various topical and parenteral antimicrobial constituents. The data used to confront each model are drawn from >200 published studies obtained from searching the literature. These studies all provide ventilator-associated pneumonia and blood stream infection data as the end points relevant to the models. The infection prevention interventions studied were not limited to SDD, with studies of exposures to other antimicrobials, such as antiseptic or antifungal agents or even non-antimicrobial-based methods of infection prevention [17,18,19,20,23,24,25,26,27,28,29,30,31,32]. Indeed, some studies had no infection prevention intervention under study, merely reporting the background rate of these end points in the population of interest. All such studies in this patient group provide clinical end point data relevant to the three postulated causation networks.

The study inclusion criteria are intentionally loose. While some of the studies were randomized with concurrent control design, others were non-randomized. Some studies used non-concurrent controls, and some were entirely without control groups. Few of the studies would have satisfied the rigorous inclusion criteria for a systematic review, where the objective is to minimize the effects of potential study biases towards deriving precise summary effects for specific interventions in specific patient groups against specific end points.

One disadvantage of testing postulated models ‘en bloc’ using SEM is that the model identified by AIC emerges as being merely optimal, not necessarily correct in every detail. However, an ‘en bloc’ comparison of postulated models with versus without key defining steps enables the chain of causation to be tested.

### 4.3. Pseudomonas and Candida Colonization

For illustration, the data (Figure 3 and Figure 4) from 289 published studies used to confront the SEMs (Figure 5, Figure 6 and Figure 7) in the SEM analysis for models of antibiotic prophylaxis on *Pseudomonas* and *Candida* colonization as latent variables are shown here [88]. These models use data not limited to studies of SDD, and they also use data from studies of infection prevention in this patient group using antiseptic and non-antimicrobial interventions, whether RCCT or cluster-randomized [92], and also observational studies without any specific study intervention. Notably, the incidence of infections among studies of TAP, with or without the PPAP component, is generally higher than among any other type of study.

The output from these studies are GSEM models as shown in the originating publications [88,89,90,91]. The results table for the multivariate regression derived from the GSEM analyses is complex, even for relatively simple models. For example, the colonization resistance model, with only two microbes, eight drivers, and four end points of interest, has a results table containing four intercepts, up to twenty coefficients, and two error terms, each having 95% confidence intervals for each of several candidate models. Here, for simplicity, only the coefficients for the key defining steps appearing in the final (optimal) model are shown (Table 1). These coefficients would be compared to other terms within the model and to similar terms in related models. Note that in these models, all exposures, including exposures to antifungal or topical antibiotic or protocolized parenteral antibiotic prophylaxis, are factorized where possible.

The interpretation of these models proceeds as follows. In all the SEM analyses [88,89,90,91], including the examples presented here for *Pseudomonas* and *Candida* colonization (Figure 5, Figure 6 and Figure 7), the colonization susceptibility model emerges as optimal based on this model having the lowest AIC. Here, this is displayed in Figure 7.

The coefficients within each SEM can then be examined with particular attention to those associated with the key defining steps. Within the SEM, these are presented in the multivariate regression tables [88,89,90,91]. They are intended as no more than approximate estimations compared to what might have been derived by a rigorous estimation of each step in a controlled study with optimal control of all variables, including potentially complex exposure interactions of significance to that isolated step.

The key defining step in the COGO model is the direct effect of TAP and antifungal exposures on *Pseudomonas* and *Candida* colonization (Figure 5). Notably, the effects of TAP on *Pseudomonas* (−0.44; −0.68 to −0.21) and *Candida* (+1.02; 0.11 to 1.91) colonization are negative and positive, respectively, in all three models. These effects are as might be expected. To this base model is then added in turn the additional defining steps for the next two models.

The key defining step of the colonization resistance model is the step between CC and *Pseudomonas* colonization (Figure 6). The coefficient for this step in the optimal model is positive (CC to *Pseudomonas* colonization; +0.37; +0.06 to +0.68, *p* = 0.01), which is contrary to the negative coefficient that would be expected from the original colonization resistance studies in mice [9,10].

The key defining step of the colonization susceptibility model is the influence of *Candida* on *Pseudomonas* colonization (Figure 7), with this coefficient (+0.37; +0.26 to +0.49) being positive.

Surprisingly, in all three models, exposure to PPAP emerges as a promoter of *Pseudomonas* bacteremia [0.73; 0.1 to 1.36], which is a finding with precedent [93,94].

For the models with other bacterial colonization, as expected, antibiotic prophylaxis has negative effects on colonization with *Pseudomonas aeruginosa*, *Acinetobacter*, and *Staphylococcus aureus* and positive effects for colonization with *Candida*. The effect of CC and the effect of the *Candida* colonization latent variable on the other bacterial colonization latent variables are positive in all models (Table 1).

It is possible, using the coefficients derived in this model, to generate predictions for *Pseudomonas* and *Candida* blood stream infections in response to exposures to the various factors as identified in the optimal model. Surprisingly, the positive effect of concurrency and *Candida* colonization on *Pseudomonas* blood stream infections is similar in magnitude to the negative effect of TAP. Also, nystatin is relatively ineffective against *Candida* colonization in the model. This relative ineffectiveness of nystatin may account for the failure of an SDD regimen containing nystatin as the antifungal in a recent large trial [38].

### 4.4. Limitations

There are multiple caveats and limitations to SEM [81,82,83,84]. The advantages, limitations, and underlying concepts of SEM methods for applications within infectious disease and infection control applications have been described [6,13].

The latent variables and the coefficients derived within SEM should only be considered as indicative, as they cannot be directly measured. They have no counterpart at the level of any one patient or patient group. The colonization resistance and colonization susceptibility models have latent variables which represent the functional state of specific microbes within the colonizing flora at some unspecified location. This is not equivalent to the colonization incidence, which is commonly measured in ICU studies as rectal colonization. These latent variables represent a propensity to facilitate invasive infections in a deliberately simplistic model. The various antibiotic-based, antiseptic, and antifungal prophylactic regimens vary with respect to intensity and duration of application and the route of administration. Specific regimens have not been individually modelled. It is a deliberate simplification to consider these as similar within each category. Other complexities not factored in are the impact of rebound on SDD withdrawal and the impact of SDD on antimicrobial resistance, which might be considered group-level rather than individual-level outcomes. Moreover, the interrelation with mortality risk and the exposure to antimicrobials for therapeutic, as opposed to protocolized prophylactic, use are not modelled [83,95,96,97,98,99].

In these examples, the included studies have been selected using intentionally broad inclusion criteria. There is considerable heterogeneity in the interventions, populations, study quality, and study designs among studies published over several decades. However, in this way, the postulated model becomes generalizable to related interventions and populations as broadly encountered in the literature, rather than specific to any one population. This heterogeneity could be seen as a strength.

## 5. Future Applications

Related opportunities for SEM include understanding the interrelation between various antimicrobials and the microbiome, which could include the prevention of blood stream infections and mortality within febrile neutropenic patients, for which there are many studies reporting clinical data [100,101]. SEM could also be applied to estimating the complex interactions between endotoxemia and Gram-negative bacteremia [102,103] and interactions between parasitic infestations [104], and modelling the effect of antimicrobial and other exposures on the occurrence of *Clostridium difficile* infection in the hospital setting [104,105,106,107].

## 6. Conclusions

Research questions such as which of three competing hypotheses best accounts for the relationship between exposure of the microbiome to various antibiotic regimens and the risk of blood stream infections are possible to explore using SEM techniques. Through modelling these postulated networks of causation-inferred mediation by multiple and potentially compound exogenous drivers operating through latent variables on multiple simultaneously observed end points, a comparison is possible. It would be impractical to study these networks, which would typically not be observable or testable within any single study in isolation. These methods rely on the availability of clinically derived data in large amounts from different sources, with which the candidate models are confronted. Surprisingly, exposing ICU patients to topical antibiotic prophylaxis emerges as harmful in this context through its effects mediated by concurrency and *Candida* colonization.

## Figures and Tables

**Figure 1 antibiotics-13-01096-f001:**
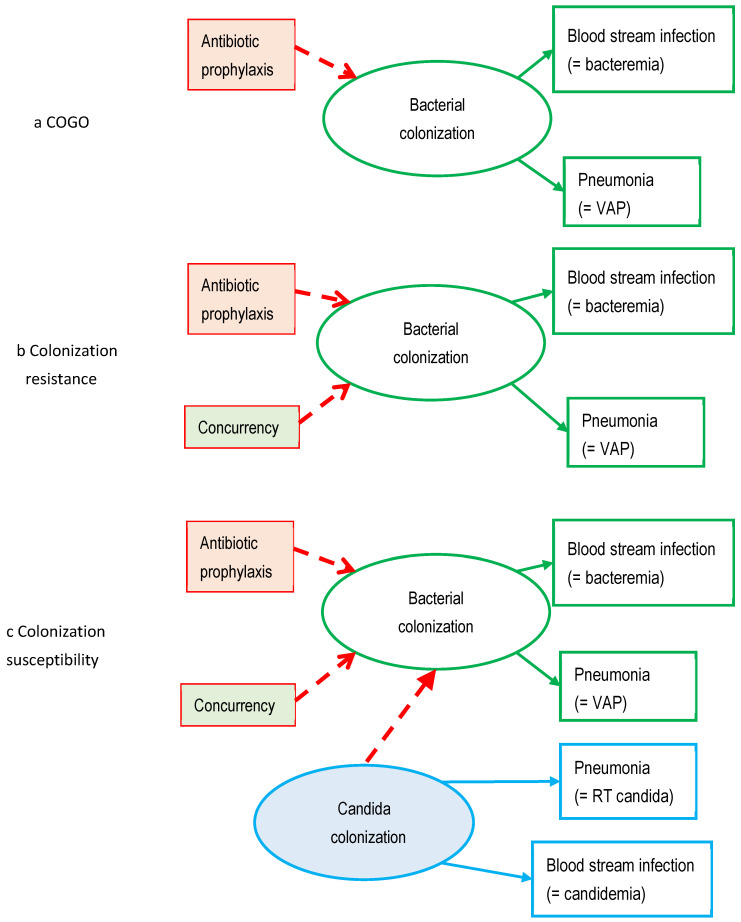
Three competing theoretical models of how exposing the microbiome (bacterial and *Candida* colonization) to topical antibiotic prophylaxis impacts the risk of blood stream and other infections. (**a**) Control of gut overgrowth (COGO), (**b**) colonization resistance, and (**c**) colonization susceptibility models. ‘Concurrency’ refers to the control and intervention groups concurrent within the same ICU. Bacterial and *Candida* colonization, being not easily measurable, are represented in the models as latent variables.

**Figure 2 antibiotics-13-01096-f002:**
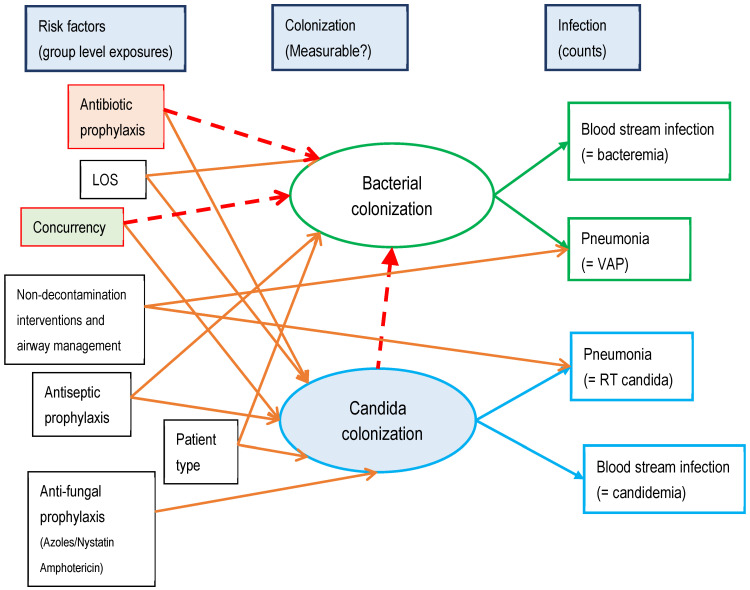
Three competing theoretical models of topical antibiotic prophylaxis mediating bacterial colonization causing blood stream and other infections are incorporated step by step into a sequence of structural equation models. Bacterial and *Candida* colonization are not easily measurable and are represented in the models as latent variables (ovals). The broken red arrows are the key defining steps for the COGO, colonization resistance, and colonization susceptibility models, respectively. ‘Concurrency’ refers to the control and intervention groups concurrent within the same ICU. Unbroken arrows are common to all models. Patient type refers to ICUs that have selective patient entry (e.g., trauma). *Candida* is not a recognized cause of ventilator-associated pneumonia (VAP), and hence here the *Candida* isolates are counted among respiratory tract (RT) isolates. LOS is group mean length of ICU stay. Model selection is based on Akaike information criteria (AIC).

**Figure 3 antibiotics-13-01096-f003:**
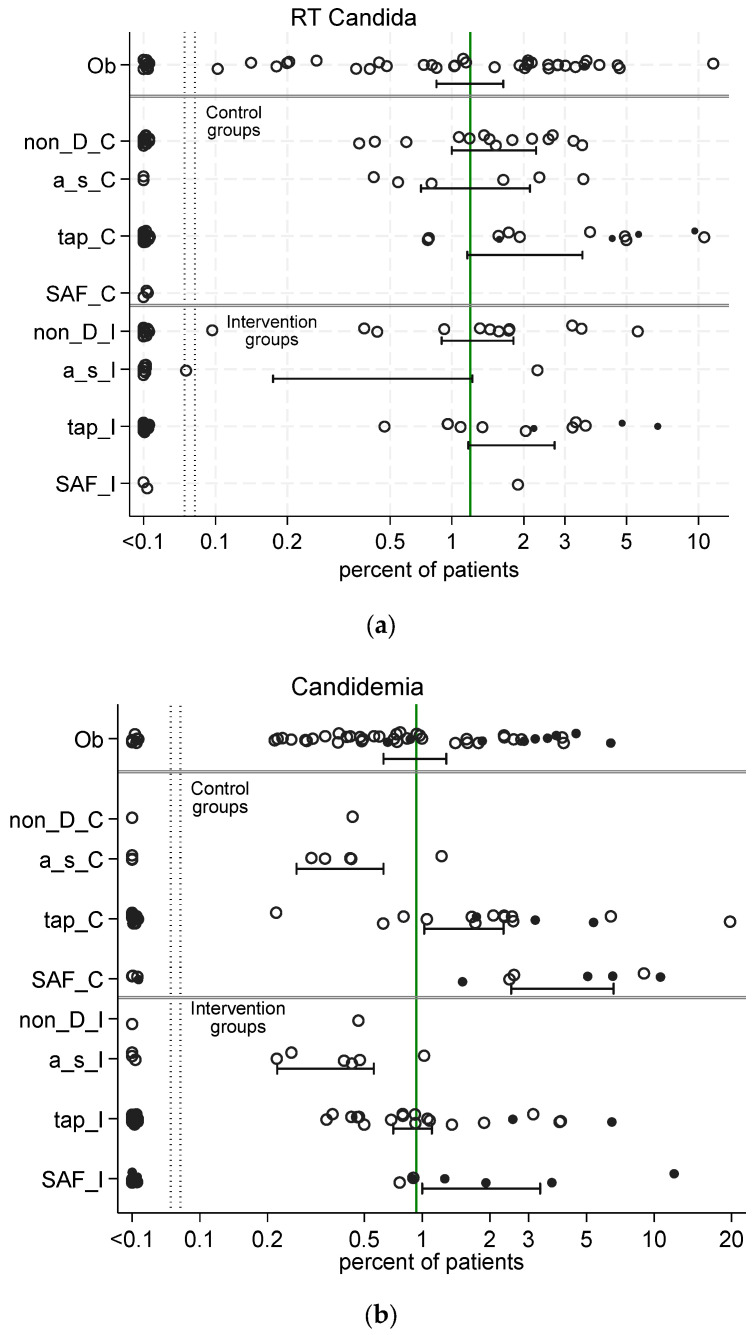
Scatter plots (logit scale) and 95% CI of respiratory tract (RT) *Candida* incidence (**a**) and candidemia (**b**) in component (control and intervention) groups of studies of various methods of infection prevention and observational studies in the ICU. Data from 289 studies as listed in reference [87]. The mean proportion (and 95% CI) derived by random-effect meta-analysis for each category of component (observational [Ob], control [_C], and intervention [_I]) group derived from observational [Ob], non-decontamination (non-D), antiseptic (a_s), topical antibiotic prophylaxis (tap), and single antifungal (SAF) studies, is displayed. The benchmark incidence in each plot is the summary mean derived from the observation studies (central vertical line). The group-wide presence of candidemia risk factors (CRF) is identified by solid symbols versus not (open). The data in the figure are listed in reference [88].

**Figure 4 antibiotics-13-01096-f004:**
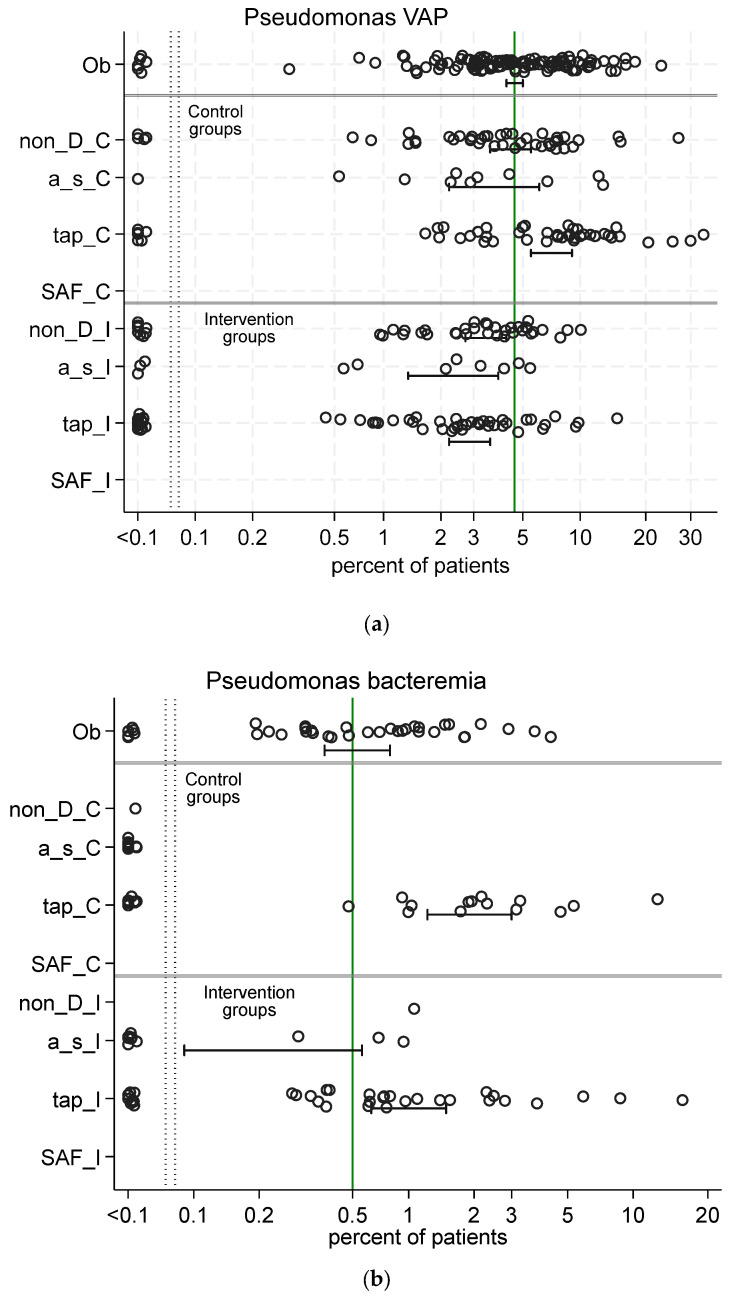
Scatter plots (logit scale) and 95% CI of *Pseudomonas* VAP incidence (**a**) and *Pseudomonas* bacteremia (**b**) in component (control and intervention) groups of various methods of infection prevention in the ICU. The benchmark incidence in each plot is the summary mean derived from the observation studies (central vertical line). Abbreviations as for Figure 3. The data in the figure are listed in reference [88].

**Figure 5 antibiotics-13-01096-f005:**
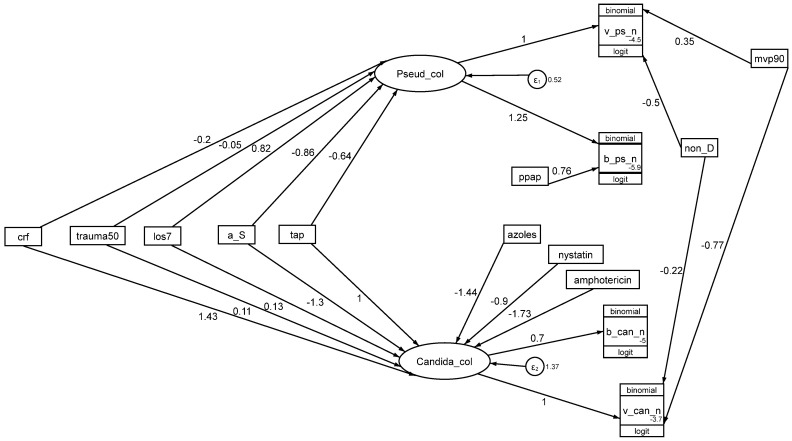
A model of COGO as a GSEM. *Candida*_col and *Pseudomonas*_col (ovals) are latent variables representing *Candida* and *Pseudomonas* colonization, respectively. The variables in rectangles are binary predictor variables representing the group-level exposure to the following: a trauma ICU setting (trauma50), mean or median length of ICU stay >7 days (los7), exposure to a topical antiseptic-based prevention method (a_S), exposure to a TAP-based prevention method (tap), exposure to a non-decontamination-based prevention method (non-D), use of mechanical ventilation more for than 90% of the group (mvp90) or exposure to PPAP (ppap), and exposure to azole/nystatin of amphotericin as antifungal prophylaxis. Groups with patient selection based on candidemia risk factors are factored (crf). The circles contain error terms (ε) associated with the latent variables. The three-part boxes represent the count data for *Candida* and *Pseudomonas* VAP (v_can_n, v_ps_n) and bacteremia (b_can_n, b_ps_n), each of which is logit-transformed with the total number of patients in each group as the denominator, using the logit link function in the generalized model of the GSEM. The Akaike information criterion (AIC) is 3974. The figure is adapted from reference [88] and used here under the terms of the Creative Commons Attribution 4.0 International License (http://creativecommons.org/licenses/by/4.0/) (accessed on 12 November 2024).

**Figure 6 antibiotics-13-01096-f006:**
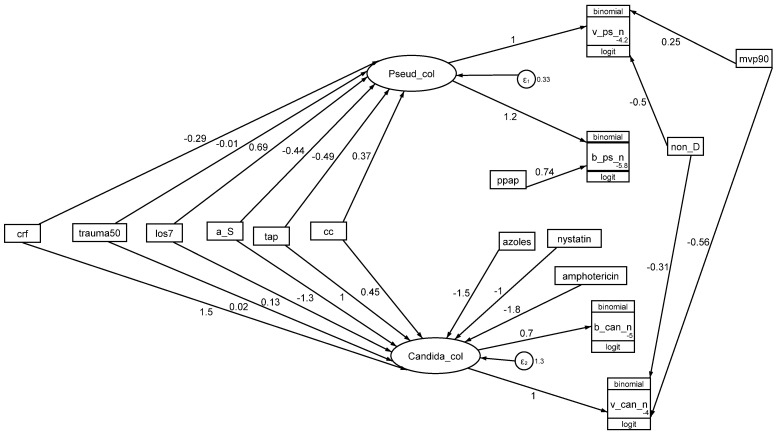
A model of colonization resistance as a GSEM. The model is as for Figure 5 but includes concurrency (CC) with a group exposed to TAP as a factor. The AIC is 3928. The figure is adapted from reference [88] and used here under the terms of the Creative Commons Attribution 4.0 International License (http://creativecommons.org/licenses/by/4.0/) (accessed on 12 November 2024).

**Figure 7 antibiotics-13-01096-f007:**
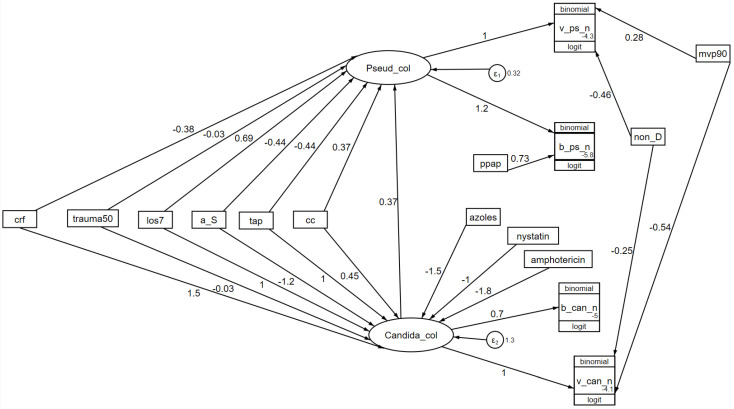
A model of colonization susceptibility as a GSEM. The model is as for Figure 6 but includes an interaction between the latent variables representing *Candida* and *Pseudomonas* colonization. The AIC is 3921. The figure is adapted from reference [88] and used here under the terms of the Creative Commons Attribution 4.0 International License (http://creativecommons.org/licenses/by/4.0/) (accessed on 12 November 2024).

**Table 1 antibiotics-13-01096-t001:** Coefficients for key defining steps derived from the optimal GSEM models ^a,b^.

Model [Ref]		COGO		Colonization Resistance		Colonization Susceptibility	
	n/N ^c^	Antibiotic Prophylaxis ^d^ → Bacterial Colonization		Concurrency ^e^ → Bacterial Colonization		*Candida* Colonization → Bacterial Colonization ^f^	
			95%CI		95%CI		95%CI
*Candida* [88]	464/279	+1.0 *	0.11 to 1.09	+0.45	−0.19 to 1.09	NR	
*Pseudomonas* [88] ^g^	464/279	−0.44 ***	−0.68 to −0.21	+0.37 **	+0.06 to 0.68	+0.37 **	0.26 to 0.49
*Acinetobacter* [89] ^h^	334/213	−0.43	−1.1 to 0.0	+0.42	−0.22 to 1.22	+0.32 *	0.01 to 0.5
Enterococcal [90] ^i^	450/274	+0.51 **	0.12 to 0.89	+0.5	−0.05 to 1.05	+0.56 ***	0.33 to 0.79
CNS [90] ^j^	450/274	+0.90 ***	0.46 to 1.33	+0.45	−0.11 to 1.01	+0.68 ***	0.34 to 1.0
*S. aureus* [91] ^k^	473/288	−0.41 **	−0.7 to −0.12	+0.4 *	0.02 to 0.72	+0.37 ***	0.25 to 0.49

Footnotes: ^a^ Legend: * *p* < 0.05; ** *p* < 0.01; *** *p* < 0.001; NR, not reported. ^b^ Coefficients for factors relating to the key defining steps within each model. For simplicity, additional factors within each model are not shown. In each case, the optimal model as indicated by AIC that emerged from confrontation with the data is the colonization susceptibility model. ^c^ n is the number of groups and N is the number of studies. ^d^ Shown in this table are all models toward developing the optimal model as indicated by AIC, which in each case was the colonization susceptibility model. ^e^ Antibiotic prophylaxis refers specifically to topical antibiotic prophylaxis. ^f^ *Candida* colonization refers to the impact of *Candida* manifest as a latent variable in the model. ^g^ *Pseudomonas* colonization (*Pseudomonas* col in Figure 5, Figure 6 and Figure 7) is a latent variable. ^h^ *Acinetobacter* colonization is a latent variable. ^i^ Enterococcal colonization is a latent variable. ^j^ Coagulase-negative Staphylococcus (CNS) colonization is a latent variable. k. *S. aureus* colonization is a latent variable.

## Data Availability

The datasets analyzed during the current study are provided in the online appendix to reference [87].

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
