# Peer review of "Structural Equation Modelling as a Proof-of-Concept Tool for Mediation Mechanisms Between Topical Antibiotic Prophylaxis and Six Types of Blood Stream Infection Among ICU Patients"

_antibiotics, 2024, doi:10.3390/antibiotics13111096_

Round 1

Reviewer 1 Report

Comments and Suggestions for Authors

1. With this being a review, I found it hard to follow the logical flow and goals of this review. It seems as though the authors should start in the context of SEM. The authors jump into SEM being used to address if antibiotic exposures mediate bacterial colonization causing bloodstream infections, and they speak at length about colonization resistance, overgrowth, colonization susceptibility first....I feel although introducing the problem is relevant it seems out of order...  It seems they should first introduce 1. What is SEM? 2. How does it work? 3. In what other contexts has it previously been used? 4. then introduce the competing hypotheses antibiotic prophylaxis mediating bacterial colonization causing blood stream problem of  5. Why is SEM an appropriate model or a superior model to address this problem versus other mediation models, what are the advantages/disadvantages to use it in this scenario? 6. NOW review how it can be applied to address if antibiotic exposures mediate bacterial colonization causing bloodstream infections and what has been done in the literature.

2. The authors talk about "predicting" BSI, SEM are not predictive models... they estimate mediation effects and analyze relationships between latent and observed variables and can test mediation hypotheses to variables, mediators, and outcomes... 

3. The goal or point of lines 379 to 444 is muddled. It seems like Figures 3 and 4 represent data from one study.., but then it seems as though Fig 5,6,7 culminate data from 4 different studies? But they refer to 289 published studies being represented in these figures (Line 388)? Are you comparing the findings in these models? Are you culminating the information from each to make some generalized conclusions? If the later, it is unclear what we are supposed to take away from this section of the review. It seems a rehashing of results from refs 87-90 with no strong take aways from the data as presented. 

5. How does this review add to the field or improve upon your recent reviews and papers on a similar topic, the figures seem quite similar 

Hurley JC. How to apply structural equation modelling to infectious diseases concepts. Clin 570 Microbiol Infect. 2022 ;28(12):1567-71  

ames C Hurley, Structural equation modelling the impact of antimicrobials on the human microbiome. Colonization resistance versus colonization susceptibility as case studies, Journal of Antimicrobial Chemotherapy, Volume 78, Issue 2, February 2023, Pages 328–337,

Hurley JC. Structural equation modelling the relationship between anti-fungal prophylaxis and Pseudomonas bacteremia in ICU patients. Intensive Care Med Exp. 2022 Jan 21;10(1):2. doi: 10.1186/s40635-022-00429-8.

Author Response

  1. With this being a review, I found it hard to follow the logical flow and goals of this review. It seems as though the authors should start in the context of SEM. The authors jump into SEM being used to address if antibiotic exposures mediate bacterial colonization causing bloodstream infections, and they speak at length about colonization resistance, overgrowth, colonization susceptibility first....I feel although introducing the problem is relevant it seems out of order...  It seems they should first introduce 1. What is SEM? 2. How does it work? 3. In what other contexts has it previously been used? 4. then introduce the competing hypotheses antibiotic prophylaxis mediating bacterial colonization causing blood stream problem of  5. Why is SEM an appropriate model or a superior model to address this problem versus other mediation models, what are the advantages/disadvantages to use it in this scenario? 6. NOW review how it can be applied to address if antibiotic exposures mediate bacterial colonization causing bloodstream infections and what has been done in the literature.

Author response… I have clarified in response to the comments to attempt to reach a balance in introducing the various concepts.

  1. The authors talk about "predicting" BSI, SEM are not predictive models... they estimate mediation effects and analyze relationships between latent and observed variables and can test mediation hypotheses to variables, mediators, and outcomes... 

Author response…I disagree. All model are predictive models although I agree with this this reviewer that SEM have an additional ability to estimate mediation effects and analyze relationships between latent and observed variables. I have attempted to clarify the text in response.

  1. The goal or point of lines 379 to 444 is muddled. It seems like Figures 3 and 4 represent data from one study.., but then it seems as though Fig 5,6,7 culminate data from 4 different studies? But they refer to 289 published studies being represented in these figures (Line 388)? Are you comparing the findings in these models? Are you culminating the information from each to make some generalized conclusions? If the later, it is unclear what we are supposed to take away from this section of the review. It seems a rehashing of results from refs 87-90 with no strong take aways from the data as presented. 

Author response… I have attempted to clarify this text in response.

  1. How does this review add to the field or improve upon your recent reviews and papers on a similar topic, the figures seem quite similar 

Author response…The figures are there as examples to compare and contrast the similarity across the related studies.

Reviewer 2 Report

Comments and Suggestions for Authors

1.       What is the novelty of this work when this model is already known? Please discuss and justify.

2.       Line 15 to 19. Confusing, long sentence, please edit or rewrite.

3.       Most of the figures are adopted from earlier published articles. There is no novelty of new interpretation out in the current manuscript. Please justify.

4.       What are the future implications of this work?

5.       Effect on gut and blood microbiome is completely a distinct scenario. How did the authors looking at these two different scenario at the same time? What are the correlations?

6.       Authors should have provide or design their own illustrative figures on the basis of published literature rather than just acquiring.

7.       What are the major blood or gut microbiota targets that considered in the model presented?

Comments on the Quality of English Language

This is fine for me. 

Author Response

  1. What is the novelty of this work when this model is already known? Please discuss and justify.

Author response…The text has been clarified.

  1. Line 15 to 19. Confusing, long sentence, please edit or rewrite.

Author response…This has been rewritten

  1. Most of the figures are adopted from earlier published articles. There is no novelty of new interpretation out in the current manuscript. Please justify.

Author response… The figures are there as examples to compare and contrast the similarity across the related studies.

  1. What are the future implications of this work?

Author response…The text has been clarified.

  1. Effect on gut and blood microbiome is completely a distinct scenario. How did the authors looking at these two different scenario at the same time? What are the correlations?

Author response…The ‘gut and blood microbiome ‘ has been removed and replaced.

  1. Authors should have provide or design their own illustrative figures on the basis of published literature rather than just acquiring.

Author response…Figure 1 is new

  1. What are the major blood or gut microbiota targets that considered in the model presented?

Author response…Figure 1 is presented to identify the relevant latent variables most relevant to the models

Reviewer 3 Report

Comments and Suggestions for Authors

The manuscript entitled  “Structural equation modelling as a tool for exploring the effect of antibiotics on the gut/blood microbiome" takes important issue in the era of alarm pathogens with high epidemic potential, it presents a very interesting approach to the problem of exposure to colonization by different species, mainly bacteria and fungi of the genus Candida, depending on the composition of the microbiome. The paper presents a rich review of the literature on microbiome analysis in the context of the potential assessment of colonization and subsequent risk of infection by different species of microorganisms.  The manuscript  is written in a very interesting way, and the rich analysis of the data lends credence to the importance of analyzing the composition of the microbiome in the context of the factor protecting the patient from a serious invasive infection during a hospital stay. I strongly recommend to publish this manuscript.

Minor editorial correction is needed, including:

Line 279: “similar” should be written by capital letter,

Author Response

The manuscript entitled  “Structural equation modelling as a tool for exploring the effect of antibiotics on the gut/blood microbiome" takes important issue in the era of alarm pathogens with high epidemic potential, it presents a very interesting approach to the problem of exposure to colonization by different species, mainly bacteria and fungi of the genus Candida, depending on the composition of the microbiome. The paper presents a rich review of the literature on microbiome analysis in the context of the potential assessment of colonization and subsequent risk of infection by different species of microorganisms.  The manuscript  is written in a very interesting way, and the rich analysis of the data lends credence to the importance of analyzing the composition of the microbiome in the context of the factor protecting the patient from a serious invasive infection during a hospital stay. I strongly recommend to publish this manuscript.

Minor editorial correction is needed, including:

Line 279: “similar” should be written by capital letter,

Author response:.... I thank the reviewer for these comments and I have attended to the correction as outlined.

Round 2

Reviewer 1 Report

Comments and Suggestions for Authors

1. Prophylactic and empirical antibiotics are primarily given systemically (IV) or orally to neutropenic fever patients as well as ICU patients, not topically. It is difficult to tell if the data pulled from the 280+ studies was censored for only topical antibiotic administration b/c of the data being reiterated throughout different publications. The title seems it might be a misnomer as it is more likely patients had systemic and oral antibiotic exposures. Although they consider TAP (topical antibiotic prophylaxis), I'm not sure its singular contribution is studied? This needs to be clarified.

2. Did they separate the route of administration of the drugs? Selective decontamination is done systemically or orally. It seems as though this would be important to separate as some drugs are metabolized differently. For example, oral vancomycin is not absorbed through the gut into the bloodstream. It appears the route of administration and antibiotic would be important to actually affect the bloodstream. How is this considered in the model? If they indeed they are only looking at TAP, it seems there needs to be more of an introduction to what TAP actually is, and the antibiotics used in TAP as most ID individuals would think of systemic antibiotics given in ICU pts, not topical. For example, all of the references referring to SDD discussed 8-11, 14, 16, 17-20 etc are referring to systemic and oral antibiotic administration (enteral and parenteral admin), not topical. So I'm not sure where the authors came up with the concept of topical antibiotic prophylaxis for SDD???  Topical antibiotics would likely only be used in skin and soft tissue infection (SSTI), burn victims, prevention in surgical wounds, impetigo, etc infections, not common ICU events as discussed herein. 

3. Moreover, no mechanisms are actually tested or shown as the title suggests.  

4. Line 365 seems unfinished, number not inserted.

5. Line 400, which studies represent only TAP versus systemically or orally administered prophylaxis? This is not clear from the references presented

6. Species and genera names should be italicized (line 425 and other places).

7. Its hard to determine what the actual research questions or goal of the paper was...  As written I am still trying to figure out of the goal of the paper was to compare the 3 competing hypotheses (line 549-551) of 1. colonization resistance, 2. overgrowth, and 3. colonization susceptibility. If that is the case, it appears some discussion of performance needs to validate which model is true. If it is thought that all events contribute, or some combination there of and that is why you look at all three, then perhaps they should not be competing models , but contributing models. 

8. If the goal of your paper is lines 34-37, then perhaps discuss each model in the context of each question, what does each model say for each question? 

9. It seems the bottom line is prophylaxis is harmful in the context of ICU bloodstream infection. I think this is a great bottom like, but perhaps this should be highlighted and supported better as it was not clear until the very last sentence. 

10. Since the outcome is only candida or pseudomonas VAP or BSI, the title should likely reflect this (and the conclusions) as this may not be true for all BSIs/other etiological agents. 

Comments on the Quality of English Language

Currently the structure (not necessarily the language used) of the paper is difficult to follow. With what questions are presented, the methods and results used to answer each question, and the conclusion for each question. 

Author Response

I thank the reviewers for their comments on the second round. These have again been most helpful.

Reviewer 1

  1. Prophylactic and empirical antibiotics are primarily given systemically (IV) or orally to neutropenic fever patients as well as ICU patients, not topically. It is difficult to tell if the data pulled from the 280+ studies was censored for only topical antibiotic administration b/c of the data being reiterated throughout different publications. The title seems it might be a misnomer as it is more likely patients had systemic and oral antibiotic exposures. Although they consider TAP (topical antibiotic prophylaxis), I'm not sure its singular contribution is studied? This needs to be clarified.

Author response…Whilst the reviewer is generally correct with respect of “Prophylactic …antibiotics..are primarily given systemically (IV) or orally to neutropenic fever patients as well as ICU patients, not topically..”, in the case of the studies here, prophylactic and antibiotics are primarily given topically to the ICU patients as SDD, which is a regimen of topical antibiotic which may [SDD] or may not [SOD] be supplement by 4 days of systemically (IV) antibiotic use. This regimen is no longer used for neutropenic patients [as discussed]. The models estimate the activities of both components [systemic and topical] separately. This point is clarified in the manuscript in lines 141-161.

  1. Did they separate the route of administration of the drugs? Selective decontamination is done systemically or orally. It seems as though this would be important to separate as some drugs are metabolized differently. For example, oral vancomycin is not absorbed through the gut into the bloodstream. It appears the route of administration and antibiotic would be important to actually affect the bloodstream. How is this considered in the model? If they indeed they are only looking at TAP, it seems there needs to be more of an introduction to what TAP actually is, and the antibiotics used in TAP as most ID individuals would think of systemic antibiotics given in ICU pts, not topical. For example, all of the references referring to SDD discussed 8-11, 14, 16, 17-20 etc are referring to systemic and oral antibiotic administration (enteral and parenteral admin), not topical. So I'm not sure where the authors came up with the concept of topical antibiotic prophylaxis for SDD???  Topical antibiotics would likely only be used in skin and soft tissue infection (SSTI), burn victims, prevention in surgical wounds, impetigo, etc infections, not common ICU events as discussed herein. 

Author response… The reviewer is correct in that the practice of SDD using topical antibiotic prophylaxis is somewhat idiosyncratic to some European ICU’s [where it has been extensively studied]. The use is based on a concept that topical antibiotics [applied to the gut] prevent infections [as summarized in the two Cochrane reviews of this topic ref 18 & 19]. The mechanism remains unclear. This use of topical antibiotics serves as a natural experiment which is explored here using SEM.

  1. Moreover, no mechanisms are actually tested or shown as the title suggests. 

Author response… I respectfully disagree with the reviewer here. The mechanisms tested are 1. colonization resistance, 2. overgrowth, and 3. colonization susceptibility [as in figure 1].

  1. Line 365 seems unfinished, number not inserted.

Author response…thankyou for spotting, this error has been corrected.

  1. Line 400, which studies represent only TAP versus systemically or orally administered prophylaxis? This is not clear from the references presented
    Author response…
  2. Species and genera names should be italicized (line 425 and other places).

Author response… thankyou for spotting, this error has been corrected.

  1. Its hard to determine what the actual research questions or goal of the paper was...  As written I am still trying to figure out of the goal of the paper was to compare the 3 competing hypotheses (line 549-551) of 1. colonization resistance, 2. overgrowth, and 3. colonization susceptibility. If that is the case, it appears some discussion of performance needs to validate which model is true. If it is thought that all events contribute, or some combination there of and that is why you look at all three, then perhaps they should not be competing models , but contributing models. 

Author response… The manuscript is a review article of a method (SEM) which has novel application to the study of possible mechanisms underlying the mediation of topical antibiotic prophylaxis. The results as given are given of examples arising in studies presented elsewhere.  The reviewer correctly identifies that the goal of the paper was to compare the 3 competing hypotheses (line 549-551) of 1. colonization resistance, 2. overgrowth, and 3. colonization susceptibility. These are conceptually competing models which share elements. The title and abstract have been changed in response to this comment.

  1. If the goal of your paper is lines 34-37, then perhaps discuss each model in the context of each question, what does each model say for each question? 

Author response…Thankyou for this comment. The questions have been addressed over the course of the manuscript.

  1. It seems the bottom line is prophylaxis is harmful in the context of ICU bloodstream infection. I think this is a great bottom like, but perhaps this should be highlighted and supported better as it was not clear until the very last sentence. 

Author response…Whilst prophylaxis might be harmful in the context of ICU bloodstream infection, this is not the main objective of this manuscript which is to review a method (SEM) which has novel application to the study of possible mechanisms here. The abstract has been changed in response to this comment.

  1. Since the outcome is only candida or pseudomonas VAP or BSI, the title should likely reflect this (and the conclusions) as this may not be true for all BSIs/other etiological agents. 

Author response…Whilst candida or pseudomonas VAP or BSI are discussed as the main example, the corresponding conclusions for the other six BSI types are summarized in Table 2 and the related discussion.

Comments on the Quality of English Language

Currently the structure (not necessarily the language used) of the paper is difficult to follow. With what questions are presented, the methods and results used to answer each question, and the conclusion for each question. 

Author response…The manuscript is a review article of a method (SEM) which has novel application to the study of possible mechanisms underlying the mediation of topical antibiotic prophylaxis. The results as given are given of examples arising in studies presented elsewhere.  

Reviewer 2 Report

Comments and Suggestions for Authors

The author successfully responded to the reviewer's comments and updated the manuscript. 

Author Response

Comments and Suggestions for Authors

The author successfully responded to the reviewer's comments and updated the manuscript. 

Author response…Thankyou for these comments